# Improved High-Quality Genome Assembly and Annotation of Pineapple (*Ananas comosus*) Cultivar MD2 Revealed Extensive Haplotype Diversity and Diversified FRS/FRF Gene Family

**DOI:** 10.3390/genes13010052

**Published:** 2021-12-24

**Authors:** Ashley G. Yow, Hamed Bostan, Raúl Castanera, Valentino Ruggieri, Molla F. Mengist, Julien Curaba, Roberto Young, Nicholas Gillitt, Massimo Iorizzo

**Affiliations:** 1Department of Horticultural Science, North Carolina State University, Raleigh, NC 27695, USA; agyow@ncsu.edu; 2Plants for Human Health Institute, North Carolina State University, Kannapolis, NC 28081, USA; hbostan@ncsu.edu (H.B.); mmengis@ncsu.edu (M.F.M.); jbcuraba@ncsu.edu (J.C.); 3Centre for Research in Agricultural Genomics CSIC-IRTA-UAB-UB, Campus UAB, 08193 Barcelona, Spain; raul.castanera@cragenomica.es; 4Biomeets Consulting, Carrer d’Àlaba 61, 08005 Barcelona, Spain; valentino.ruggieri@biomeets.es; 5Research Department of Dole, Standard Fruit de Honduras, Zona Mazapan, La Ceiba 31101, Honduras; roberto.young@dole.com; 6Core Genomics Lab, David H. Murdock Research Institute, Kannapolis, NC 28081, USA; nick@berkleyrd.com

**Keywords:** pineapple MD2, *Ananas comosus*, improved assembly, haplotype diversity, FHY3/FAR1 genes, FRS/FRF genes

## Abstract

Pineapple (*Ananas comosus* (L.) Merr.) is the second most important tropical fruit crop globally, and ‘MD2’ is the most important cultivated variety. A high-quality genome is important for molecular-based breeding, but available pineapple genomes still have some quality limitations. Here, PacBio and Hi-C data were used to develop a new high-quality MD2 assembly and gene prediction. Compared to the previous MD2 assembly, major improvements included a 26.6-fold increase in contig N50 length, phased chromosomes, and >6000 new genes. The new MD2 assembly also included 161.6 Mb additional sequences and >3000 extra genes compared to the F153 genome. Over 48% of the predicted genes harbored potential deleterious mutations, indicating that the high level of heterozygosity in this species contributes to maintaining functional alleles. The genome was used to characterize the FAR1-RELATED SEQUENCE (FRS) genes that were expanded in pineapple and rice. Transposed and dispersed duplications contributed to expanding the numbers of these genes in the pineapple lineage. Several AcFRS genes were differentially expressed among tissue-types and stages of flower development, suggesting that their expansion contributed to evolving specialized functions in reproductive tissues. The new MD2 assembly will serve as a new reference for genetic and genomic studies in pineapple.

## 1. Introduction

Pineapple is the second most important tropical fruit globally. The fruit, which contains nutritionally valuable vitamins and bioactive enzymes (e.g., bromelain), is consumed in both fresh and processed forms. Worldwide production was estimated at over 30 million tons in 2019 (http://www.fao.org/, accessed on 23 March 2021). The consumption of pineapple fruit and the use of pineapple-derived supplements has been steadily increasing over the past decade and continues to increase each year [1]. ‘MD2’ is the most widely-grown fresh fruit market cultivar by major production companies due to its consistently large fruit size and better fruit quality as compared to other cultivars [2].

Developing new cultivars with improved characteristics is key to overcoming production challenges (e.g., disease or abiotic stress) and responding to changes in market demand. The use of genomic resources helps to accelerate the process of cultivar development, allowing breeders to meet consumer and grower needs more quickly than with traditional breeding approaches alone [3]. Incorporation of advanced genetic and genomic resources into a breeding program provides several advantages, including increased efficiency when breeding for specific traits and reduced screening time for determining the presence/absence of those traits in a breeding population [3].

Pineapple is a highly heterozygous diploid species with 25 chromosomes and an estimated haploid genome size of 563 Mb [4]. Multiple genome assemblies for pineapple representing two cultivars (MD2 and F153) and a close relative species (*Ananas bracteatus*, CB5) are currently available. However, the quality of these genomes in terms of sequence contiguity (contig N50) is relatively low (MD2 v1 = 57 kb). The MD2 v1 genome that represents the most commonly grown pineapple variety is not anchored at the chromosome level, and only 56% of the estimated genome size is anchored at the chromosome level for the F153 genome [2,5]. Homologous chromosomes have not been fully phased for any MD2 pineapple genome to date. New long-read sequencing technologies, such as PacBio, allow for the iterative improvement of genome assemblies for economically important crops, such as pineapple [6]. Genomic interaction data, such as Hi-C, facilitate the assembly of high-quality scaffolds and the reconstruction of haplotype phases [7,8]. Also, gene predictions obtained from long-read sequencing technology, such as PacBio Iso-Seq, allow researchers to obtain full-length transcripts, increasing the reliability of the predictions [9].

To continue building on recent advances in pineapple genetics and genomics, here we present a high-quality, phased, chromosome-scale assembly of MD2. Comparative haplotype analysis was performed to study the potential impact of heterozygosity on allele diversity. Comparison with previous pineapple and other crop genomes was performed to highlight assembly and gene prediction improvements, potential chromosome rearrangements, and to characterize regulatory and resistance genes, including the genes that code for the transcription factor (TF) FAR-RED ELONGATED HYPOCOTYL3 (FHY3) and its homolog FAR-RED-IMPAIRED RESPONSE1 (FAR1), as well as FAR1-RELATED SEQUENCE (FRS) and FRS-RELATED FACTOR (FRF). These TFs make up the FRS/FRF family, which has been implicated in multiple developmental functions including chloroplast division and chlorophyll biosynthesis, circadian clock regulation, flowering time regulation, starch biosynthesis, and the biotic and abiotic stress responses [10,11,12,13]. The results of this study provide novel resources for genetic and genomic analyses in pineapple that are critical for advancing modern breeding strategies in this crop.

## 2. Materials and Methods

### 2.1. Plant Material Collection, DNA and RNA Extraction, and Sequencing

A diploid, commercial fresh-fruit market variety, MD2, was used for whole-genome sequencing with Pacific Biosciences (PacBio, Menlo Park, CA, USA) and Hi-C proximity ligation. Pineapple MD2 plants were shipped to Dr. Iorizzo’s lab from Dole in La Ceiba, Honduras and grown in the greenhouse at the North Carolina Research Campus, Kannapolis, NC, USA.

For DNA extraction, young leaves were harvested and then immediately frozen in liquid nitrogen, stored at −80 °C, and freeze-dried for 24 h before use. Dried leaves were ground into powder using a Genogrinder 2000 tissue homogenizer (SPEX SamplePrep, Metuchen, NJ, USA). High molecular weight genomic DNA was extracted from dry powdered leaf tissue using the CTAB method outlined in Charlotte et al. (2016) [14]. The DNA quantity and purity were determined using a Qubit fluorometer (Invitrogen, Carlsbad, CA, USA) and NanoDrop spectrophotometer (Thermo Fisher Scientific, Waltham, MA, USA), respectively. DNA was also evaluated on a 0.8% agarose gel and Agilent 4200 TapeStation instrument (Agilent Technologies, Santa Clara, CA, USA) to determine the fragment size distribution.

For RNA extraction, fresh tissues from the leaf, meristem, stem, root, flower, and fruit of greenhouse-grown MD2 pineapples were harvested and ground in liquid nitrogen. Total RNA was extracted from collected tissues using the Sigma Spectrum Total RNA kit (Sigma-Aldrich, St. Louis, MO, USA). RNA integrity was evaluated on a 1% agarose gel and Agilent 2100 Bioanalyzer instrument. RNA was quantified using Qubit and the purity was tested using NanoDrop.

A PacBio genomic library for MD2 was prepared using the SMRTbell Express Template Prep Kit (Cat. #101-357-000) following the protocol for preparing >15 kb libraries (PN 101-397-100) (Pacific Biosciences, Menlo Park, CA, USA). The SMRTbell library was analyzed on the Agilent 4200 TapeStation instrument to determine if size-selection was necessary prior to sequencing with the PacBio Sequel system at the David H. Murdock Research Institute (DHMRI, Kannapolis, NC, USA). PacBio Iso-Seq libraries were prepared with the SMRTbell Express Template Prep Kit 2.0 (Cat. #100-938-900) according to the Iso-Seq Express Template Preparation for Sequel and Sequel II Systems protocol (PN 101-763-800). For Iso-Seq library preparation, RNA from leaf, root, fruit, and flower tissues were maintained separately to prepare one library each, while RNA from meristem and stem tissues were combined in equimolar amounts to prepare one combined library. A total of five libraries were prepared and each library was sequenced on a SMRT cell with the PacBio Sequel system. The Hi-C library for MD2 was prepared using the Phase Genomics Proximo Hi-C Plant Kit (Seattle, WA, USA). Quantity and quality checks were performed by Phase Genomics, including qPCR analysis, electrophoretic assay on the Agilent 2100 Bioanalyzer instrument, and spike-in control. Hi-C libraries were sequenced with the Illumina HiSeq 2500 using 150 bp paired-end (2 × 150 bp) run chemistry.

The PacBio genomic reads were processed for error correction and to generate consensus sequences using the FALCON pipeline [15]. Iso-Seq reads were further processed with IsoSeq3 pipeline (https://github.com/PacificBiosciences/IsoSeq (accessed on 29 January 2019), with the ccs considering -min-rq 0.9, lima considering --isoseq --dump-clips --no-pbi --peek-guess, and refine considering --require-polya and cluster), generating circular consensus sequence (CCS) reads for subsequent data analysis. The Illumina Hi-C sequences were quality-checked with FastQC (https://www.bioinformatics.babraham.ac.uk/projects/fastqc/, accessed on 23 May 2019) and cleaned using Trimmomatic [16] (parameters: ILLUMINACLIP:2:30:10:2:TRUE SLIDINGWINDOW:10:30 LEADING:30 TRAILING:30 HEADCROP:10 MINLEN:45), which removed adapter sequences. PCR duplicates were flagged for removal using SAMBLASTER [17]. After this step, cleaned sequences were retained for downstream analysis.

### 2.2. Genome Assembly, Phasing, and Scaffolding

To generate an improved MD2 pineapple genome assembly, newly generated PacBio long reads and Hi-C data were integrated. The new MD2 v2 genome assembly presented here was generated using a de novo assembly approach, and four published pineapple genome assemblies (MD2 v1, F153 v3, F153 v7 and CB5) [2,5,18] were used only for comparative analysis.

A de novo assembly for MD2 was generated using a FALCON assembler [15]. FALCON-UNZIP was later used to reconstruct the un-phased contigs (haplotigs) from the initially assembled contigs [15]. In addition to the sequence error correction implemented in the FALCON pipeline, Pilon v.1.24 [19] was used to further polish the assembled sequences using the Hi-C Illumina short reads. Hi-C data were subsequently incorporated for further phasing of the genome using FALCON-Phase [7], resulting in a fully-phased genome assembly (phases P0 and P1).

Scaffolding of the MD2 v2 genome assembly was performed by Phase Genomics using the Proximo pipeline (https://phasegenomics.com/products/proximo/, accessed on 26 August 2019). The haplotype-phased contigs (haplotigs) obtained from FALCON-Phase were used as the input for scaffolding with Proximo, to create a chromosome-scale genome assembly. A linkage map including 46,860 SSR markers was used to support clustering of contigs into pseudomolecules during the scaffolding analysis [20]. SSR marker sequences developed for pineapple variety F153 were obtained from the Pineapple Genomics Database [20] and aligned against the MD2 v2 FALCON-Phase genome assembly using BWA-aln (default parameters) [5,20]. Note that only the physical coordinates of each SSR marker mapped in the F153 genome were publicly available and not the genetic distance (cM).

The Hi-C interaction heat map in conjunction with the published SSR linkage map were used to identify and correct chimeric regions following two conditions: (1) non-collinearity with the order of the markers in the SSR linkage map, and (2) the presence of gaps in the Hi-C interaction along the heatmap diagonal. The positions of these chimeric regions were corrected to maximize the Hi-C interaction signals along the diagonal. After this correction process, the F153 SSR linkage map aligned with high collinearity to the 25 chromosomes of the MD2 v2 assembly.

In both phases, the longest 25 scaffolds represented the 25 pineapple chromosomes. The P0 and P1 assemblies were used as input files for a final step of error correction and gap filling.

### 2.3. Assembly Error Correction and Gap-Filling

Assembly errors were manually corrected in Juicebox (v.1.8.8) [21]. Misassemblies and incorrect sequence orientations were detected in the phased, scaffolded MD2 v2 assembly by examining the Hi-C heatmap for zones of depleted genomic interaction (no red signals along diagonal) and by performing a collinearity analysis with the linkage map as described above. The genomic positions of the F153 SSR marker sequences in the MD2 v2 scaffolds were compared to their genomic positions in the F153 v3 reference genome. The Hi-C interaction heatmap provides a visual guide for assessing assembly quality. The red color along the diagonal indicates points of interaction between neighboring genomic regions. Off-diagonal red indicates: (1) highly repetitive genomic regions, such as centromeres or telomeres, or (2) misassembled/chimeric sequences. When performing manual corrections, in those cases where the collinearity analyses with the F153 SSR linkage map and Hi-C interaction heatmap were not in agreement for a region, preference was given to the Hi-C data. The rationale behind this criterion was based on the fact that the Hi-C data represent the MD2 genome while the linkage map represents a different genetic background, and it cannot be excluded that the differences between the linkage map and the MD2 genome represent true structural differences.

The first round of manual corrections was based on the collinearity between the F153 SSR linkage map (using marker locations of the F153 v3 genome) and the scaffolded MD2 v2 genome. Assembly error corrections were only kept if they both improved the collinearity with the SSR map and did not negatively affect the Hi-C interaction heatmap. The second round of manual corrections was based solely on the Hi-C interaction heatmap visualization. These first and second rounds of manual corrections were conducted on each phase separately. Both phases were aligned to each other using the Nucmer module of MUMmer4 [22] to identify regions of non-collinearity between individual haplotype assemblies. Non-collinear regions, including differences in the order and orientation of contigs between phases, were corrected based on the linkage map collinearity and the Hi-C interaction heatmap as described above. After manual correction of the MD2 v2 genome assembly, LR_GapCloser [23] was run with three iterations for gap-filling using the cleaned genomic PacBio reads.

### 2.4. Genome Quality Analysis

Assembly quality was determined using several metrics that assess correctness, contiguity, and completeness. Assembly correctness was determined using the Hi-C heatmap, F153 SSR linkage map collinearity, and by performing a contamination check against fungal, bacterial, and viral genomes with NCBI BLASTn. Contiguity was determined based on the contig and scaffold N50 lengths, the percentage of contigs anchored to chromosomes, and the ratio of the overall number of contigs assembled to total assembly length. Completeness was determined based on the overall percentage of estimated genome size assembled, long terminal repeat (LTR) assembly index (LAI) score [24], and by using Benchmarking Universal Single-Copy Orthologs (BUSCO) v.3 [25] against the embryophyta_odb10. The completeness of the assembly gene space was assessed using the percentage of mapped transcriptome reads and the BUSCO score. Published transcriptomic data for pineapple were obtained from the NCBI SRA database (project accession nos. PRJNA648819, PRJNA648693, PRJNA494788, PRJNA393610, PRJNA356904, PRJNA331052, PRJNA310033, PRJNA305042, PRJNA237705), and sequences were cleaned using Trimmomatic [16] and quality-checked with FastQC. Transcripts were aligned to the MD2 v2 assembly P0 and P1 haplotypes separately using STAR [26] (--outSAMstrandField intronMotif --outSAMattrIHstart 0 --outFilterMismatchNmax 2 --outSAMtype BAM SortedByCoordinate).

The MD2 v2 assembly was aligned against a previously sequenced pineapple genome (F153 v7) and one representing a wild relative *A. bracteatus* (CB5) [5,18]. Both of these assemblies were assembled at chromosome level. Alignment of the genomes was performed using the Nucmer module of MUMmer4 (default parameters) and visualization of the alignments was performed using the web-based tool, Dot (https://dnanexus.github.io/dot/, accessed on 19 March 2020).

### 2.5. Genome Annotation

Gene models were predicted by implementing a hybrid strategy using the MAKER pipeline, a tool for annotating a reference genome using empirical and ab initio gene prediction deploying AUGUSTUS v.2.5.5 [27] and SNAP [28], as well as the GeMoMa pipeline [29,30], a homology-based gene prediction program that predicts gene models in target species based on gene models in evolutionary-related reference species. These processes involved several iterations of homology-based and in silico gene prediction steps, using high-throughput short- and long-read sequences as well as gene models obtained from multiple closely related species and model organisms (*A. comosus* [5], *Arabidopsis thaliana* [31], *Carica papaya* [32], *Musa acuminata* [33], *Oryza sativa* [34], *Solanum lycopersicum* [35], *Sorghum bicolor* [36], *Vitis vinifera* [37], and *Zea mays* [38]), downloaded from Phytozome and used as the input for training.

Structures of the predicted gene models were manually verified by aligning PacBio Iso-Seq full-length transcripts using GMAP aligner (--min-identity = 0.99 --min-trimmed-coverage = 0.95 --nosplicing) [39].

The putative function of the predicted genes was annotated using public databases, including NCBI non-redundant protein, KOG, GO, and InterPro. NCBI BLASTx was used to compare the predicted coding sequences (CDS) (e-value ≤ 1 × 10^−10^) with the non-redundant protein database (downloaded in December 2020). Blast2GO v.1.4.11 was used to annotate the GO terms of genes with default parameters. The protein domains were annotated using the Blast2GO InterProScan module [40] based on all available protein databases.

Eukaryotic clustering of orthologous group (KOG) analysis was performed with the eggNOG Mapper v.5.0 Blast2GO extension [41]. The eggNOG Mapper tool assigns orthologs and transfers functions to query genes using phylogenetic inference.

Reciprocal, ungapped BLASTn alignments of MD2 v2 P0 and MD2 v1 CDSs with query coverage and percent identity parameters set to 100 (-ungapped, -qcov_hsp_perc 100, and -perc_identity 100) revealed which genes had structural differences between the two MD2 genomes. Additionally, a reciprocal BLASTn alignment (default parameters) of MD2 v2, MD2 v1, and F153 v3 CDSs indicated which genes predicted in the MD2 v2 genome were not predicted in the previous genomes.

Salmon [42] (default parameters) was used to align Illumina RNA-seq reads from the NCBI SRA database (project accession no. PRJNA305042) with the predicted genes to determine if the new genes identified by BLASTn were expressed (TPM > 0).

Disease resistance genes (R-genes) and regulatory genes were predicted to assess how the new MD2 v2 genome affected the prediction of some known important gene families. Transcription factors (TFs), transcriptional regulators (TRs), and chromatin regulators (CRs) were identified using PlantTFcat [43] on pineapple MD2 v2, MD2 v1, and F153 v3 gene models, as well as for various other species. R-genes were identified using DRAGO2 [44] for the same set of gene models/species. The number of genes in each gene family was compared between MD2 v2 and the published pineapple genomes to determine the number of extra TFs, TRs, CRs and R-genes that were identified in this study.

The EDTA pipeline [45] was used to obtain a non-redundant catalog of TE families using MD2 v2 P0. This step included running LTRharvest [46], LTR_FINDER_parallel [47], and LTR_retriever [48] for detecting LTR-retrotransposons, GRF [49] and TIR-Learner [50] for detecting TIR transposons, and HelitronScanner [51] for detecting Helitrons. Finally, RepeatModeler (http://www.repeatmasker.org/RepeatModeler/, accessed on 20 March 2020) was used to complement the library with non-LTR retrotransposons and other TE families bypassed by the previous tools.

RepeatMasker (http://www.repeatmasker.org, accessed on 3 April 2020) was used to annotate the two MD v2 haplotypes, as well as the MD2 v1 [2], CB5 [18], and F153 v7 [18] genomes using our TE library (Appendix A). In parallel, intact LTR-retrotransposons, TIRs, and Helitrons were identified in the four genomes by running the corresponding *ltr*, *tir*, and *helitron* modules of the EDTA pipeline.

Then, a Perl script (parseRM.pl -p option, available at https://github.com/4ureliek/Parsing-RepeatMasker-Outputs, accessed on 9 April 2020) was used to parse the raw alignment outputs from RepeatMasker and get a detailed summary for each repeat family and class as well as the total amount of repeats per genome. LTR-assembly index (LAI) [24] and LTR-retrotransposon insertion age were calculated using LTR-retriever and compared across the four genomes to provide additional support for the high quality of the MD2 v2 genome assembly.

### 2.6. Haplotype Comparison

Alignment-based comparison of the phased haplotypes of the MD2 pineapple assembly was conducted using the pipeline recently developed for the phased diploid potato genome [52]. Syntenic regions between the P0 and P1 haplotypes were identified and plotted using MCScanX [53]. Homologous chromosomes were aligned using MUMmer v.4.0 and structural variants (≥100 bp) were detected from differences reported by the *show-diff* function. Presence and absence variation (PAV) genes were identified and defined as genes that lacked a homolog on the complementary haplotype, while its surrounding genes had homologs that were collinear in position between the two haplotypes. SNPs and indels between haplotypes were annotated using SnpEff [54].

### 2.7. Identification and Characterization of FRS/FRF Transcription Factors

TFs identified with PlantTFcat were compared across multiple monocot and dicot species to determine if any contraction or expansion of specific gene families occurred in pineapple. Based on the prediction of TFs in pineapple and other genomes, and in pineapple and rice, both members of the order Poales within the monocot lineage had a larger number of genes belonging to the FRS/FRF family (see Section 3.6). Therefore, this gene family was characterized in terms of the mode of duplication, shared ancestry, and pattern of expression across different tissue and flower developmental stages.

Ortholog prediction of FRS/FRF genes from pineapple and other species was performed using OrthoMCL v.2.0.9 (https://orthomcl.org, accessed on 7 January 2021) [55]. To explore the evolutionary relationships of FRS/FRF gene family members in pineapple, the FRS/FRF amino acid sequences from pineapple and other species were used. Multiple sequence alignments of all FRS/FRF proteins were performed by using MUSCLE (https://www.ebi.ac.uk/Tools/msa/muscle/, accessed on 17 August 2021) [56], with default parameters. Subsequently, phylogenetic trees were constructed using MEGA-X v.10.2.6 software (http://www.megasoftware.net, accessed on 17 August 2021) [57] via the maximum-likelihood (ML) method with the following parameters: node robustness was detected using the bootstrap method, and the bootstrap was set to 100 replications.

The tissue-specific expression, level of expression (measured as FPKM), and differential expression between tissues were evaluated with RSEM [58] using RNA-seq data from NCBI Bioprojects 483249 and 656750. An AcFRS gene was considered not to be expressed if it had FPKM <2. AcFRS genes with FPKM ≥2 were considered to be expressed and AcFRS genes with FPKM ≥10 were considered “highly” expressed. NCBI Bioprojects 483249 and 483249 were used for differential expression analysis (up- or downregulated genes). For RNA-seq data analysis and interpretation of the results, each Bioproject was treated as an independent experiment. NCBI Bioproject 483249 is comprised of 11 different pineapple tissues representing the transcriptome during fruit development, and NCBI Bioproject 656750 is comprised of 27 different pineapple floral tissues representing the transcriptome during flowering.

## 3. Results and Discussion

### 3.1. Genome Assembly and Quality Assessment

In total, 32.6 Gb of raw PacBio whole-genome sequences, 52 Gb of Hi-C interaction data, and 143.7 Gb of Pacbio Iso-Seq sequences were generated (Appendix A). Error correction and consensus generation of the raw PacBio genomic reads resulted in 2,014,848 (~25 Gb) high-quality error-corrected consensus sequences. A total of 125,054 (~201 Mb) polished, high-quality full-length circular consensus sequence (CCS) Iso-Seq reads were generated for subsequent data analysis (Appendix A). After trimming the Illumina Hi-C reads and removing PCR duplicates, 49 Gb of cleaned sequences were retained for downstream analysis.

PacBio sequences were used to perform de novo assembly and Hi-C sequences along with a linkage map were used to scaffold and phase the assembly [8], correct chimeric regions, and anchor the assembly to chromosomes (Appendix A). Gap-filling of the assembly with LR_Gapcloser resulted in ~63 kb additional known sequence added (Appendix A). The final phased assembly spanned 1.075 Gb, including 543.5 Mb for P0 and 531.6 Mb for P1 (Table 1 and Appendix A), accounting for ~96.5% (P0) of the estimated genome size (563 Mb per haploid phase) [4]. Each phase of the assembly contained 63 scaffolds and contigs, and 99.7% of the P0 and P1 assembled sequences were anchored to 50 chromosomes (25 chromosomes for each phase) (Figure 1a).

The F153 SSR linkage map aligned with high collinearity to the 25 chromosomes, demonstrating correct ordering and orientation of the final MD2 v2 assembly (Appendix A). Similarly, the Hi-C heatmap showed a uniform distribution of genomic interactions along the diagonal, demonstrating the proximity of the assembled sequences and the quality of the assembly (Figure 1b and Appendix A). The overall N50 in P0 and P1 was >22 Mb and the contig N50 was >1.5 Mb, similar to those of other high-quality genome assemblies, such as *Actinidia chinensis* v3.0 and *S. bicolor* v3.0 [36,59]. The longest chromosome (Chr 1) spanned 43.5 Mb and the shortest (Chr 25) spanned 4.4 Mb (Appendix A). The length of the longest contig was >5.9 Mb, covering a large part of the chromosome 13 long arm.

Gene space in the MD v2 assembly was assessed using transcriptome sequences and BUSCO analysis. Mapping 222 sets of transcriptome sequences from the NCBI SRA database indicated that 95.5% aligned with the MD2 v2 genome assembly (Appendix A). BUSCO analysis indicated that >97% conserved genes had a match in the MD v2 genome, of which >95% were detected as a complete structure. These results demonstrated that this assembly covered the majority of the gene space. Finally, no significant sequence contamination was detected by BLASTn alignment against a custom-made database that included bacterial, viral, and fungal DNA sequences.

### 3.2. Genome Assembly Comparisons

Comparison of all the currently available *Ananas* sp. genomes highlighted major improvements in the MD2 v2 assembly and some interspecific chromosomal rearrangements.

Compared to the previous MD2 v1 assembly, the MD2 v2 assembly had a 26.6-fold increase in contig N50 length, 19.4 Mb of total additional sequence (including Ns), and 33.5 Mb of additional known sequence at the contig level (Table 2). Most importantly, the MD2 v2 was assembled at the chromosome level and phased, while the MD2 v1 was only assembled at the scaffold level and not phased. The MD2 v2 assembly had a 13.3-fold increase in contig N50 length over the F153 v7 genome, 161.6 Mb of total additional sequence, and 168.3 Mb of additional known sequence at the contig level (Table 2). This represented >30% of the estimated genome size.

Compared to the *A. comosus* genomes that are assembled at the chromosome scale [18], MD2 v2 had 40.1% more of its sequence anchored to chromosomes than F153 (Table 2).

At the structural level, the comparison with F153 v7 [18] indicated that the assemblies were highly collinear; however, several novel sequences appeared to be inserted into the MD2 v2 assembly (Appendix A). Over 33 Mb (76% of its length) of new known sequences were assembled in chromosome 1, and a total of 10 chromosomes in the MD2 v2 assembly had >10 Mb of additional known sequence compared to the corresponding chromosomes from F153 v7 (Appendix A). The increase in known sequence content could be partially attributed to the fact that more repetitive regions were sequenced and assembled in the MD2 v2 assembly (Figure 1a,b). Regions of high repeat density in the MD2 v2 chromosomes coincided with newly assembled sequence regions (Figure 1b). MD2 v2 chromosome 25 was the only chromosome in the assembly that was smaller than (and, therefore, had less known sequence) the corresponding chromosome in F153 v7. This difference was due to chimeric sequences identified in the F153 v7 chromosome 25. Indeed, the F153 SSR linkage map and MD2 Hi-C data supported the assembled structure of MD2 v2 chromosome 25 (Appendix A). Other potential chimeric regions and very large gaps in the F153 v7 chromosomes also existed (Appendix A). For example, the MD2 v2 chromosome 1 was collinear to F153 v7 chromosome 1 and 24, indicating that the F153 v7 chromosome 1 and 24 should be combined into one sequence (Appendix A). These results were supported by the contiguous interaction in the Hi-C heatmap for MD2 v2 chromosome 1 (Appendix A).

Comparison with the CB5 genome (*A. bracteatus*) [18] revealed moderate collinearity with multiple rearrangements, including a small and large inversion in CB5 chromosomes 12 and 8 (Appendix A), respectively, and large translocations in CB5 chromosomes 3, 7, 11, 19, and 20 (Figure 1b and Appendix A). To gain some preliminary support for the presence of these chromosome rearrangements, the interaction signals of the MD2 Hi-C data mapped to the CB5 genome were used for verification. Non-collinear regions of CB5 chromosomes 8 and 12 were manually inverted to establish putative collinear regions. After the manual inversions, the sequence spanning the breaking points of the inverted regions had strong Hi-C interaction signals outside of the diagonal, indicating that these regions were more proximal, consistent with the original CB5 assembly (Appendix A). These results suggested that the rearrangements represented true interspecific chromosomal differences between the CB5 and MD2 genomes.

### 3.3. Repetitive Sequence Annotation and Analysis

TEs accounted for about 62% of the MD2 v2 pineapple genome (P0 and P1) (Appendix A). The two phases slightly differed, with the sequence masked 337.17 Mb for P0 and 329.34 Mb for P1. On average, the MD2 v2 class I elements (LTR, LINE, and SINE) and class II elements (TIR, MITE, and Helitron) occupied 211.85 and 121.4 Mb DNA sequences, accounting for 39.41% and 22.58% of the genome, respectively. Among LTR-TE, Ty3/Gypsy and Ty1/Copia represented approximately 24.94% and 6.72% of the genome, respectively.

Compared to the previous MD2 v1 assembly, the MD2 v2 has about 22.61 Mb additional sequence annotated as TE, with most of this difference being due to an increase (1.76%) in Ty3/Gypsy LTR elements (Figure 1). Wider differences were instead observed in comparisons with the other pineapple genomes considered, showing an increase of about 4% and 10% over CB5 and F153 v7, respectively (Appendix A).

Up to 12,630 and 12,439 intact TEs were annotated in the MD2 v2 P0 and P1 haplotypes, respectively, in contrast to the 5721 annotated in the previous MD2 v1 genome (Figure 2a). This increment was found in LTR, TIR, MITE, and Helitron, but it was especially evident in the LTR order. These elements occupied up to 61.9% of the MD2 v2 genome and 59.11% in the MD2 v1 genome. The LTR-assembly index (raw LAI) of MD2 v2 ranged from 21.29 (P0) to 21.77 (P1), whereas for MD2 v1 it was 6.63 (Figure 2b), indicating a much higher proportion of intact vs. total LTR-retrotransposons in the MD2 v2 genome, the result of an improved assembly. The LAI score was evenly high across all the chromosomes (Figure 1a). The corrected LAI index of MD2 v2 (19.66, P1) was comparable or superior to other high-quality plant reference genomes (i.e., *Arabidopsis* TAIR10: 14.9, rice MSU7: 21.1, and maize B73 v4: 20.7 [24]). In comparison to the CB5 and F153 v7 genomes, MD2 v2 also showed a significantly higher LAI and proportion of intact elements (Figure 2a,b).

Analysis of the LTR-retrotransposon insertion time in MD2 v2 showed a strong peak of activity at 200 Ky and a very low number of old elements (Figure 2c). A similar profile was found in the CB5 genome, and to a lower extent, in MD2 v1. F153 v7 lacked recent LTR-retrotransposon peaks and had a peak of activity at approximately 1.6 My. The distribution of elements per age suggested that the younger elements, likely forming large regions of repetitive sequences with high similarity, could not be assembled in the F153 v7, but were assembled in the MD2 V2 assembly.

### 3.4. Gene Prediction and Annotation

In total, 60,141 gene models for the diploid-phased MD2 v2 genome were predicted; 30,591 and 29,550 in P0 and P1, respectively. This represented over 6000 and 3000 extra genes compared with the MD2 v1 and F153 genomes, respectively. Overall, the general statistics (e.g., gene length, intron and exon length) of the predicted MD2 v2 gene models were in line with previously predicted gene sets for pineapple MD2 v1 and F153 v3, as well as gene sets for eight non-pineapple species (Appendix A). The same set of predicted genes was reported for both F153 v3 [5] and F153 v7 [18]; therefore, one gene prediction was used for comparisons of MD2 v2 and F153.

In the MD2 v2 assembly, the structures of 15,275 and 13,655 genes were supported by PacBio Iso-Seq transcript alignment in the P0 and P1 haplotypes, respectively. BUSCO analysis of MD2 indicated that the v2 predicted gene set had an 11.8% and 4% higher rate of completeness compared to v1 and F153 v3, respectively and the least number of fragmented genes (Appendix A).

BLASTn alignments of MD2 v2 P0 and MD2 v1 CDSs revealed that only 3612 genes had an identical structure, while the majority of the predicted genes had a different structure. Iso-Seq data supported the quality of the MD2 v2 gene prediction (Appendix A). Additionally, reciprocal BLASTn alignment of MD2 v1 and v2 CDS indicated that a total of 5689 genes predicted in the v2 genome were not predicted in the v1 genome. Also, a total of 2060 genes predicted in the MD2 v2 genome were not predicted in the F153 v3 genome and 1483 genes had no BLAST hits to genes in either the MD2 v1 or F153 v3 genomes; therefore, these were designated as novel genes in pineapple.

Alignment of Illumina RNA-seq reads from the NCBI SRA database with the full set of predicted genes showed that 911 of the new genes were expressed, providing additional evidence for the functional relevance of these novel pineapple genes.

In MD2 v2 P0, 19,552 (64%) genes were fully annotated with computationally reliable GO terms, 2709 (9%) genes were assigned a putative GO term, and 6384 (21%) genes had BLAST hits, but no associated GO terms. Overall, only 1946 (6%) genes had no functional annotation assignments. In the MD2 v2 P1 haplotype, 19,832 (67%) genes were annotated with GO terms and 2412 (8%) genes had a putative GO assignment. One hundred and fifty (1%) genes had protein domain hits to the InterProScan database, but did not have any BLAST hits to the non-redundant protein database. Finally, 5729 (19%) genes had BLAST hits but no associated GO terms and 1427 (5%) genes had no functional annotation assignments.

Overall, 26,982 KOGs were obtained from the analysis. A total of 25,368 (83%) genes were assigned to at least one KOG functional category based on orthologous proteins (Appendix A), which means that some of the annotated genes were assigned to more than one orthologous group. A large portion (83%) of the predicted genes were successfully assigned a function based on orthologous group clustering, providing additional evidence for the new predictions.

Forty-eight percent of the novel MD2 v2 genes were annotated with at least one of the databases used (Appendix A) and the distribution of KOG functional categories assigned to the novel genes (Appendix A) was similar to that of the complete set of predicted genes, except the set of novel genes had a higher proportion of sequences associated with DNA replication, recombination, and repair functions. Direct GO counts of the novel genes indicated that they function in a diverse array of biological processes, including DNA integration and repair, telomere maintenance, protease activity, and flowering (Appendix A).

The pineapple MD2 v2 P0 assembly encoded 3550 regulatory genes (TF, TR, and CR) and 1467 R-genes (Appendix A). The MD2 v2 predicted gene set encoded 686 and 662 additional regulatory genes (TF, TR, and CR) and 118 and 211 additional R-genes compared to former pineapple gene sets MD2 v1 and F153 v3, respectively.

### 3.5. Haplotype Comparison

To provide insight into the divergence between the two haplotypes, we estimated polymorphisms between the 25 homologous chromosome pairs. Based on the alignment of the genes on the two haplotypes, 185 syntenic blocks were identified. Between homologous chromosomes, 13,367,404 SNPs, 1,998,259 INDELs, and 12,468 structural variants (SVs, >50 bp) were identified (Figure 3 and Appendix A; Appendix A). Among the SVs identified, 548 SVs spanned >10 kb. Comparison of the two haplotypes exhibited an intragenomic diversity ranging from 1.1% on chromosome 6 to 4.5% on chromosome 18 (Figure 3 and Appendix A). Based on synteny and annotation, out of 60,141 predicted genes in the two haplotypes, 28,811 pairs (56,167 genes, 93% of all predicted genes) were identified as having homologs on the two haplotypes, and therefore, were considered as homologous gene pairs.

Among them, 12,745 gene pairs harbored variants with moderate to high impact on the protein coding regions, which could represent potential alternative alleles. Furthermore, a total of 3974 present and absent (PAV) genes were identified between the two haplotypes.

GO enrichment analysis for the PAV genes identified 26 GO terms including 18 biological processes, five molecular functions and three cellular components that were significantly enriched. Biological processes such as DNA integration, positive regulation of the hydrogen peroxide metabolic process, base-excision repair, and AP site formation via deaminated base removal were the most enriched GO terms (Appendix A). ADP binding, pre-mRNA 5′-splice site binding, and 5′-deoxyribose-5-phosphate lyase activity were among the molecular functions that were significantly enriched in the GO terms; these biological processes were located in cellular components such as the spindle pole body and eukaryotic translation initiation factor 3 complex. In addition to these genes with GO term annotation, 219 were also identified as R-genes. These results suggested that PAV genes may play an important role in cellular maintenance and defense.

### 3.6. Characterization of FRS/FRF Family Transcription Factors

Comparative analysis of TFs revealed that pineapple and rice harbored a larger number of FRS/FRF genes. A total of 82 pineapple MD2 v2 and 84 rice genes were identified as proteins containing the FAR1 DNA binding domain (IPR004330). Among the other genomes included in this analysis, the number of genes predicted as FRS/FRF family genes was much lower, ranging from six in banana (*M. acuminata*) to 49 in sorghum (*S. bicolor*) (Appendix A). Interestingly, the number of FRS/FRF genes was also lower in F153. It is unlikely that the larger number of genes detected in MD2 represents an expansion relative to F153 since they represent the same species and germplasm pool. Given the overall lower number of genes predicted in the F153 genome (Table 2), we readily suggest that this difference is due to misprediction in F153. Due to the large number of FRS/FRF genes in pineapple MD2 and rice, we wanted to investigate to what degree the expansion of this gene family was the result of shared duplications versus lineage-specific duplications. Also, we wanted to investigate which specific subgroups expanded their potential function by orthology, and their possible functional specialization. In the model species *Arabidopsis*, the FRS/FRF family has been implicated in multiple developmental functions related to traits of particular importance for pineapple cultivar development [5], including circadian clock regulation and flowering time regulation [10,11,12,13]. Recent work in pineapple identified FRS genes as being implicated in flower ovule development [60]. Characterizing this family of genes will lay the foundaition for future genetic studies in pineapple. To this end, the curated annotation, mode of duplication, evolutionary/phylogenetic relationships, and pattern of expression were characterized. Since the FRS/FRF family of TFs is derived from ancient transposases and contains a transposase domain, all AcFRS genes were searched against a plant TE database using BLASTn. None of the predicted AcFRSs had significant similarity with TEs. In addition, predicted AcFRS genes harbored other conserved domains including: Nuclear transcription factor Y subunit A (IPR001289), Zinc finger CCHC-type (IPR001878), WRKY domain (IPR003657), and Zinc finger PMZ-type (IPR006564). These results demonstrated that the predicted AcFRS genes were not TEs. It is also important to note that 19 AcFRS genes (23%) were PAV genes and 10 were part of the novel genes that were not predicted in the previous pineapple genome assemblies. This observation further demonstrated the value of the MD2 v2 assembly and gene prediction improvements.

The AcFRSs were grouped into one of seven groups (A–G) based on their domain content (Appendix A) and named as AcFRS1-AcFRS82. Specific DNA binding domains within a TF protein determine its potential targets for transcriptional regulation. All identified MD2 v2 FRS/FRF family genes contained at least a FAR1 DNA binding domain (IPR004330), and the majority (62) contained the MULE transposase domain (IPR018289) (Appendix A). Group G, containing genes with a FAR1 DNA binding domain (IPR004330) and Zinc finger PMZ-type domain (IPR006564), made up the majority of AcFRS genes (46, 56%). The second largest group (F, 29 genes) contained genes with only the FAR1 DNA binding domain (IPR004330). Groups A–D each contained only a single gene. Group A contained a single gene with the Nuclear transcription factor Y subunit A (IPR001289), FAR1 DNA binding (IPR004330), and Zinc finger PMZ-type (IPR006564) domains. Group B contained a single gene with four domains: Zinc finger CCHC-type (IPR001878), WRKY (IPR003657), FAR1 DNA binding (IPR004330), and Zinc finger PMZ-type (IPR006564). Group C contained a single gene with Zinc finger CCHC-type (IPR001878) and FAR1 DNA binding (IPR004330) domains. Group D contained a single gene with Zinc finger CCHC-type (IPR001878), FAR1 DNA binding (IPR004330), and Zinc finger PMZ-type (IPR006564) domains. Finally, group E contained three genes with WRKY (IPR003657), FAR1 DNA binding (IPR004330), and Zinc finger PMZ-type (IPR006564) domains. We speculate that the differences in domain structure among the AcFRS groups create a complex network of FRS/FRF family TFs with both new and overlapping functions that are capable of binding a wide range of *cis*-elements of target genes. Among the AcFRSs, group G appears to be expanded, and therefore, is more likely to contain genes that serve redundant roles; some may have acquired new functions or new targets as a result of expansion in this group.

Orthologous analysis of FRS TF proteins from all species resulted in 57 total groups (Appendix A). Twenty-three of those groups included at least one FRS genes from pineapple genomes (MD2 and F153) and nineteen groups included at least one FRS gene from rice. The majority (>65%) of the pineapple FRS genes grouped with orthologous FRS from other species, including some that have been functionally characterized. For instance, orthogroup 533 included nine AcFRSs, as well as *Arabidopsis* orthologs FHY3 and FAR1 that respond to light signals to regulate/modulate multiple processes including flowering time, seed dormancy, starch synthesis, the shade avoidance response, and balance between growth and defense responses under shade conditions [10,61]. Orthogroup 304 included eight AcFRSs and *Arabidopsis* FRS7 and FRS12, which coregulate flowering time and glucosinolate biosynthesis [62,63]. Orthogroup 756 included six AcFRSs and *Arabidopsis* FRS6 and FRS8, which have been suggested to regulate flowering time [64]. Interestingly, in these three orthogroups (304, 533, and 756), the number of AcFRS genes was higher than in the other species, suggesting that pineapple may have evolved FRS orthologs with new or specialized functions. In comparison with rice FRS (OsFRS) genes, we noted that although rice and pineapple share several FRS orthologous groups, the groups that harbor a larger number of FRS genes in each species are different (e.g., group 240: 19 AcFRS and 1 OsFRS). This demonstrated that independent duplication events contributed to accumulating the larger number of FRS genes observed within the Poales lineage.

Phylogenetic analysis clustered FRS/FRF genes into eight clades, four of which were split into two subclades (Figure 4). In this analysis, only phylogenetic clades that contained either pineapple or *Arabidopsis* FRSs were assigned a clade number (clade I–VII). Neither orthologous nor phylogenetic analyses appeared to group the AcFRS proteins based on their conserved domains. Phylogenetic clade VIB contained only pineapple-derived FRS/FRF genes and clade IIB contained primarily pineapple FRS/FRF genes (pineapple and one single papaya gene). One phylogenetic group containing *O. sativa* FRS/FRF genes (clade VB) did not contain any pineapple FRS/FRF genes. Within clades containing both pineapple and rice FRS/FRF genes, there was a clear separation between pineapple FRS/FRF genes and *O. sativa* FRS/FRF genes. These results suggest that the expansion of the FRS/FRF gene family in pineapple and rice was largely the result of independent expansion events. However, some shared signatures of conservation within this lineage were observed. For example, within clade VII, 84 out of 93 (90%) FRS/FRF genes belonged to Poales species, suggesting that the ancestor of this clade was selected within this lineage and duplicated independently after their divergence.

To examine the potential mechanism underlying the FRS/FRF family expansion in pineapple, AcFRS duplicated gene pairs were identified with DupGen_finder software [65], which classified each pair into one of five categories of duplicated gene pairs: whole genome duplication (WGD), tandem duplication (TD), proximal duplication (PD), transposed duplication (TRD), and dispersed duplication (DSD) pairs. Among these categories, the DSD category had the most duplicated gene pairs (121), followed by the TRD category (38) (Appendix A). FRS/FRF genes are derived from TEs [12]; therefore, the results of the duplication mode analysis were consistent with what would be expected for these genes. In other genomes, such as rose, it has been demonstrated that the FRS/FRF gene family likely expanded due to TE-associated DSD [66]. The DSD and TRD modes of duplication contributed to accumulating a larger number of FRS genes in pineapple, which evolved in pineapple-specific clades (IIB and VIB) and may have acquired new specialized function within the pineapple lineage. For instance, 10 AcFRSs grouped in the same subclade (IIA) as *Arabidopsis* AtFRS6 and AtFRS8, but subclade IIB has eleven additional AcFRSs with no proximal *Arabidopsis* FRS/FRF genes. AtFRS6 and AtFRS8 have been shown to negatively regulate flowering time [64]; therefore, AcFRSs in clade II may play a role in regulating the flowering time in pineapple. AcFRSs in subclade IIB may have acquired new functions related to flowering, leading to divergence from the AcFRSs in subclade IIA.

Available RNAseq data were used to evaluate the tissue-specific expression, expression level, and differential expression of AcFRS genes. This analysis revealed that 13 AcFRSs were expressed in all tissues, while 69 were expressed in at least one tissue and 13 were not expressed (Appendix A). Several AcFRSs were specifically expressed in bract, ovule, petal, sepal, and/or stamen tissue and highly expressed in the ovule, stamen, and gynoecium.

Differential expression analysis across 11 different pineapple tissues representing the transcriptome during fruit development revealed that three AcFRS genes were specifically upregulated in bract tissue and two were upregulated in both the bract and sepal (Appendix A). Differential expression analysis across 27 different floral tissues at different developmental stages identified 30 differentially expressed AcFRS genes. Of these, 18 were upregulated in one or more tissues, including a subset DE during ovule, sepal and stamen development, and 12 were downregulated, including nine during the final stages of petal development.

Overall, out of 32 unique DE genes, the majority (23, 72%) were derived from pineapple lineage-specific duplication (e.g., clade VIB), through transposed or dispersed duplication mechanisms (Figure 4). Similarly, all eight genes expressed in a tissue-specific manner were duplicated through a dispersed or transposed mechanism. These results demonstrated that expansion of AcFRS genes in the pineapple lineage contributed to evolving specialized expression patterns in reproductive tissues, which may have resulted in specialized functions related to the light-signaling pathway in floral tissues and reproductive organ development. These results complement previous findings, indicating that FRS/FRF genes in pineapple are one of the TF gene families highly expressed in flower tissue [67].

## 4. Conclusions

The final MD2 v2 assembly presented in this study represents the first MD2 chromosome-scale level assembly and the first pineapple phased genome assembly. Comparative analysis with all available pineapple genomes demonstrated higher completeness in term of sequence contiguity, gene content, and structure. Haplotype analysis, uncovered at the genomic level, the impact that the high level of heterozygosity maintained in this crop due its outcrossing nature has on genes’ content and allelic diversity. Comparative analysis with the genome of its close relative *A. bracteatus* highlighted some major genomic differentiation. Overall, the MD2 v2 genome assembly represents the most complete assembly of the pineapple cultivated germplasm. In conclusion, the MD2 v2 genome represents a new valuable resource for genetic and comparative genome analysis in pineapple that is critical for advancing marker-assisted breeding strategies in this crop.

## Figures and Tables

**Figure 1 genes-13-00052-f001:**
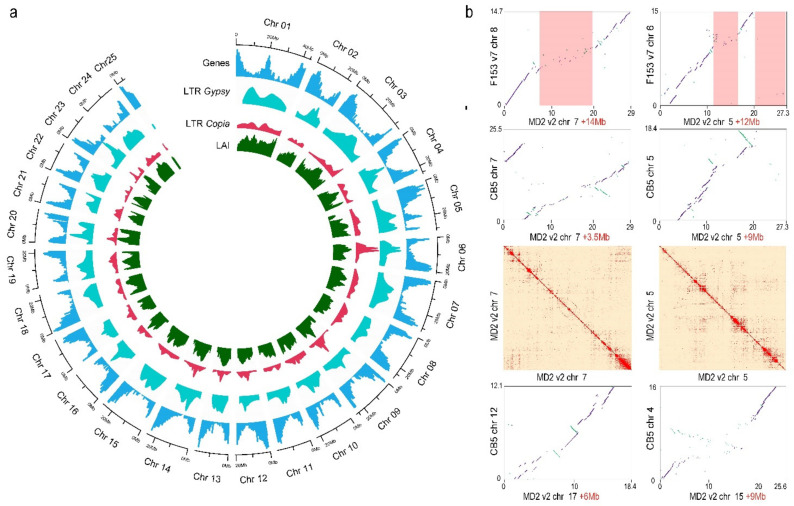
MD2 v2 P0 genomic features’ landscape and comparative analysis. (**a**) CIRCOS plot illustrating the location/density of gene models, Ty3/Gypsy LTR, Ty1/Copia LTR, and LTR Assembly Index (LAI) score along the MD2 v2 P0 assembled chromosomes; (**b**) Visual representation of the quality of the MD2 v2 genome compared to other high-quality pineapple genomes (F153 v7 and CB5). Numbers in the *x* and *y* axes represent Mb sequences. Additional sequences assembled in MD2 v2 were primarily located in regions with high amounts of repeats (i.e., centromeres, highlighted with red shaded boxes). Contiguous interaction along the Hi-C plot validates the quality of the newly assembled regions. In each chromosome comparison, the number highlighted in red indicates the additional sequences assembled in the MD2 v2 assembly.

**Figure 2 genes-13-00052-f002:**
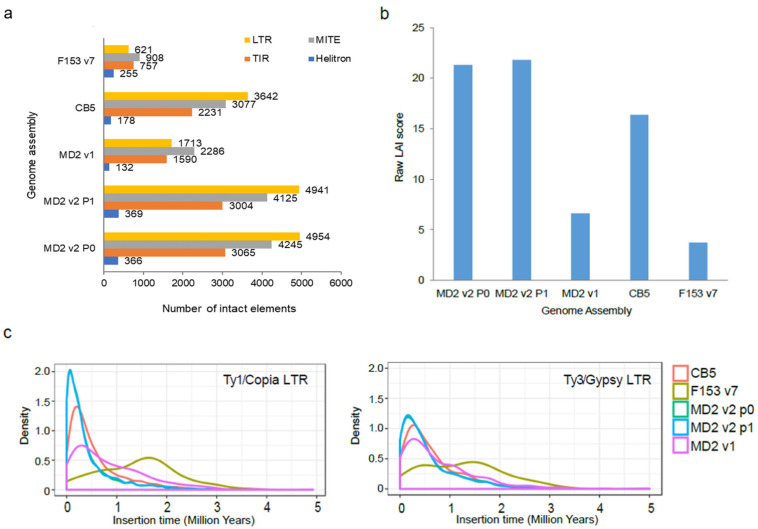
Annotation of repetitive sequences in MD2 v2 and other pineapple genomes (MD2 v1, F153 v7, and CB5). (**a**) Intact transposable elements (TEs) assembled in each pineapple genome; (**b**) Raw LTR Assembly Index (LAI) score for each pineapple genome. The LAI score is calculated as the ratio of intact elements to total elements in the assembly; therefore, a higher LAI score indicates a higher quality assembly with a larger proportion of intact elements assembled; (**c**) Insertion time for Ty3/Gypsy LTR and Ty1/Copia LTR elements in pineapple genomes. Lower insertion time indicates more recent origin. Density was calculated using the function geom_density from ggplot2 using the insertion time of all intact elements in million years.

**Figure 3 genes-13-00052-f003:**
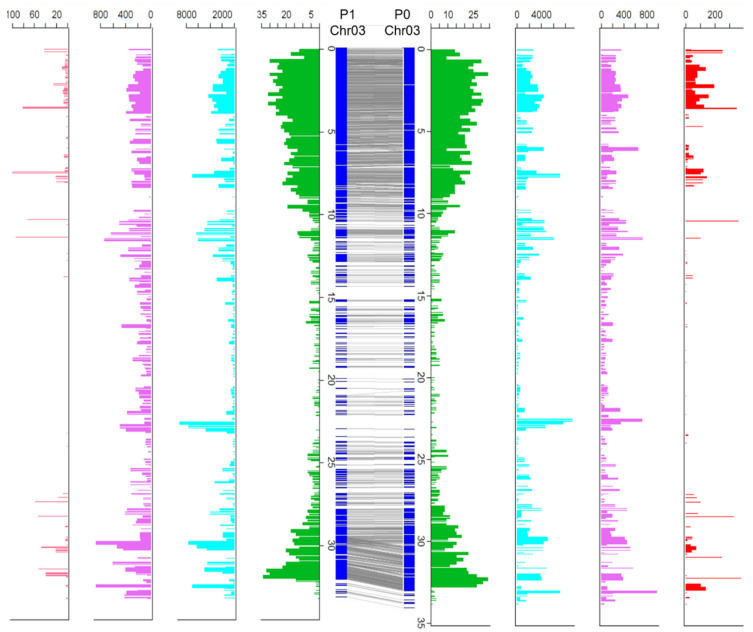
Haplotype diversity in the MD2 chromosome 3. The central blue bars represent the two haplotypes of chromosome 3. The gray lines indicate collinear genes. Outer plots include: distribution of genes (green), SNP density (cyan), INDEL density (pink), and density of potential deleterious effect variants (red). All numbers were determined in 200 kb windows.

**Figure 4 genes-13-00052-f004:**
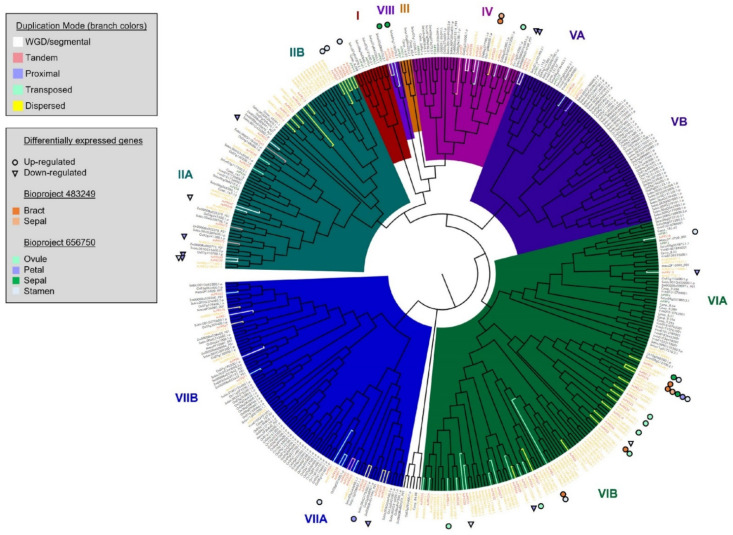
Phylogenetic tree of FRS/FRF genes from MD2 v2, other pineapple genomes (MD2 v1 and F153 v3), and eight non-pineapple species (*A. thaliana*, *C. papaya*, *M. acuminata*, *O. sativa*, *S. lycopersicum*, *S. bicolor*, *V. vinifera*, and *Z. mays*). Only clades containing FRS/FRF genes from MD2 v2 or *Arabidopsis* were numbered (Clades I–VIII). Gene names in orange correspond to pineapple MD2 v2, yellow correspond to other pineapple genomes (MD2 v1 and F153 v3), and green correspond to *A. thaliana*. Duplication mode is indicated by branch color. Differentially expressed (DE) MD2 v2 genes are indicated by colored circles or triangles next to gene names.

**Table 1 genes-13-00052-t001:** Statistics of the MD2 v2 pineapple genome assembly and gene prediction.

	MD2 v2 P0			MD2 v2 P1		
	#	Length [bp]	%	#	Length [bp]	%
Assembly feature						
Sequences	63	543,505,080	96.5 */101.0 **	63	531,615,398	94.4 */98.8 **
Contigs	812	543,433,016	96.5 */101.0 **	805	531,544,088	94.4 */98.8 **
Max. sequence length		43,498,842			42,511,760	
Min. contig length		5678			4092	
Max. contig length		5,969,083			5,971,173	
Contig N50 length		1,524,720			1,521,169	
Scaffold N50 length		21,996,178			23,016,244	
Chromosome-anchored sequence		541,772,120	99.7		529,885,665	99.7
Genome annotation						
Transposable element content		337,179,261	62.0		329,341,567	62.0
Gene models	30,591	35,382,141		29,550	34,906,836	
Genes in pseudomolecules	30,590	35,382,088	100	29,550	34,906,836	100

* Estimated genome size of 563 Mb. ** Estimated genome size of 526 Mb. # indicates number of sequences for a given metric.

**Table 2 genes-13-00052-t002:** Assembly statistical comparisons between MD2 v2 and publicly available pineapple genomes.

	MD2 v2 P0 vs. MD2 v1			MD2 v2 P0 vs. F153 v7			MD2 v2 P0 vs. CB5		
	#	Length [bp]	%/Fold Change	#	Length [bp]	%/Fold Change	#	Length [bp]	%/Fold Change
Assembly feature									
Sequences	−8385	+19,435,418	+3.5% */+3.6% **	−3070	+161,599,861	+28.7% */+30.7% **	−40	+30,269,689	+5.4% */+5.6% **
Contigs	−20,927	+33,521,821	+6.0% */+6.2% **	−8550	+168,298,756	+29.9% */32.0% **	−1158	+30,384,325	+5.4% */+5.6% **
Max. sequence length		+42,211,785	+33.8 Fold		+26,164,174	+2.5 Fold		+16,366,087	+1.6 Fold
Min. contig length		+5677	+5678 Fold		+5497	+30.4 Fold		−6456	−2.1 Fold
Max. contig length		+4,742,061	+4.9 Fold		+4,957,247	+5.9 Fold		+3,781,837	+2.7 Fold
Contig N50 length		+1,467,419	+26.6 Fold		+1,410,093	+13.3 Fold		+1,098,024	+3.6 Fold
Scaffold N50 length		+21,843,094	+143.7 Fold		+10,290,687	+1.9 Fold		+1,817,735	+1.1 Fold
Chromosome-anchored sequence *		+541,772,120	NA		+225,928,334	+40.1%		+55,160,525	+9.8%
Genome annotation									
Transposable element content		+26,531,143	+2.76%		+136,080,198	+10.03%		+38,062,886	+3.76%
Gene models	+6993	+7,217,109		+3567	+3,729,585		NA	NA	NA
Genes in pseudomolecules	NA	NA		+6698	+6,439,201		NA	NA	NA

* Estimated genome size of 563 Mb. ** Estimated genome size of 526 Mb. NA indicates cases where data for calculating differences were not available for the published genomes. # indicates number of sequences for a given metric.

## Data Availability

Data including the genome sequence, annotation, and raw sequencing reads have been deposited in the NCBI under BioProject ID PRJNA719415 and submission ID SUB9421765.

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
