# Peer review of "Improved High-Quality Genome Assembly and Annotation of Pineapple (Ananas comosus) Cultivar MD2 Revealed Extensive Haplotype Diversity and Diversified FRS/FRF Gene Family"

_genes, 2021, doi:10.3390/genes13010052_

Round 1

Reviewer 1 Report

This paper is the first MD2 chromosome level assembly and the first pineapple stepwise genome assembly, which is very important for future genomic studies of pineapple and its relatives. Furthermore, the findings on FAR1 revealed by the sequencing data in this study are very interesting. However, I think there are some minor corrections to be made and the way the data is presented, so I would like to ask for your consideration.

Comments:

1. Why did you choose chr3 as Fig.3, and why did you choose an average intragenomic diversity ranging?

2. The explanation of Fig. 4 is insufficient. The meaning of the red and green letters is not shown. It is clear that they are derived from the pineapple genome (MD2 v2) and the Arabidopsis genome, respectively. In addition, to emphasize the origin of pineapple, it would be better to color the genes of the other pineapple genome.

3. Which of the following is "clade IVB" shown in L.615?

4. The page number in the upper right corner is wrong.

5. The layout of Table 1 is a bit broken.

6. It is better to match the significant figures, e.g. P.7 L.316.

7. The term "LTR Gypsy" is not often used; " LTR of Gypsy" or "Ty3/Gypsy LTR" is often used.

8. Some abbreviations, such as TE and DE, are missing from the explanation.

9. The abbreviation "nr" is used, but in the Method (P.5 L.232-233), nr=non-redundant protein, but the Results (P.11 L.479), nr protein is used. Since it is only used twice, there is no need to use the abbreviation.

10. Only Fig. S13 is labeled "Supplementary Figure 13".

Author Response

Thank you, we appreciate the reviewer comments and edits. It helped us to improve the quality of the manuscript. We addressed all the comments and edits. Please see below a point by point response.

Reviewer 1 Comments:
Comment 1. Why did you choose chr3 as Fig.3, and why did you choose an average intragenomic diversity ranging?
Answer: No specific reason. The intention of adding the average was just to give information on intragenomic diversity in context of the whole genome instead of trying to list the numbers for each chromosome. We removed the part of the sentence that includes the average intragenomic diversity (see line 513-514)

Comment 2. The explanation of Fig. 4 is insufficient. The meaning of the red and green letters is not shown. It is clear that they are derived from the pineapple genome (MD2 v2) and the Arabidopsis genome, respectively. In addition, to emphasize the origin of pineapple, it would be better to color the genes of the other pineapple genome.
Answer: Per the suggestions of the reviewer, genes from other pineapple genomes were colored on Fig. 4 and an explanation for colored gene names was added to the figure caption (see Fig. 4)

Comment 3. Which of the following is "clade IVB" shown in L.615?
Answer: Sorry, this was a typo. Any mention of “clade IVB” has now been corrected to “clade VIB”.

Comment 4. The page number in the upper right corner is wrong.
Answer: The page numbering is now correct. However, if accepted, we will request assistance from the editorial team to modify the header format on pages 10 and 11. The issue seems to be related to presetting on the template provided by the journal.

Comment 5. The layout of Table 1 is a bit broken.
Answer: We were unable to see how the Table 1 layout is broken, it is not broken on the file that we received after revision from genes. If the reviewer would please provide a specific explanation of how the layout of Table 1 is not correct, we will address it. All tables and figures have been formatted using styles pre-configured by the journal.

Comment 6. It is better to match the significant figures, e.g. P.7 L.316.
Answer: We moved the reference to figure 1a to the next paragraph (see line 336). Also we verified that significant figures were cited in the text.

Comment 7. The term "LTR Gypsy" is not often used; " LTR of Gypsy" or "Ty3/Gypsy LTR" is often used.
Answer: All occurrences of “LTR Gypsy” and “LTR Copia” have been changed to “Ty3/Gypsy LTR” and “Ty1/Copia LTR”, respectively.

Comment 8. Some abbreviations, such as TE and DE, are missing from the explanation.
Answer: These abbreviations are now fixed in all tables/figures or their captions.

Comment 9. The abbreviation "nr" is used, but in the Method (P.5 L.232-233), nr=non-redundant protein, but the Results (P.11 L.479), nr protein is used. Since it is only used twice, there is no need to use the abbreviation.
Answer: All instances of “nr” have been changed to “non-redundant protein”. (see line 234, 235, 260, 486)

Comment 10. Only Fig. S13 is labeled "Supplementary Figure 13".
Answer: This figure has now been relabeled as Figure S13 to match the format of the other supplementary figures (see line 512, 515).

Reviewer 2 Report

The manuscript describes more complete and better annotated genome of MD2 pineapples, an agriculturally important cultivar. It also made an attempt to characterize one of gene families, which was found to have about 80 members. This was an interesting observation, however, not unique. Authors mentioned that rice also had similar number of genes, while other species varied in the number. It is not clear from the manuscript why authors decided to characterize this particular family. Looks like the only reason was the large number of genes in it. While the first part, the full genome sequencing and better annotation compared to the previous versions of pineapple genome, has clear scientific value and is informative, the second part, characterization of the FAR gene family, is unnecessary, incomplete and somewhat controversial. I would recommend a major revision.

Points to address:

Title “FAR1 gene family” should be “FHY3/FAR1 gene family” or better possibly “FRS/FRF family” as more general and newer name. On the other hand, the use of the term “expansion” for this family claimed in the title is a bit vague. Since it is not clear from the manuscript why the expansion is even claimed. Authors stated that pineapple and rice have similar numbers of FHY3/FAR1 genes (around 80), while other tested species had less (6 in banana and 49 in sorghum).

Lane 26 “Compared with other species genomes, MD2 has an expansion” is incorrectly worded comparison. It reads that MD2 is a species, while MD2 is a cultivar of pineapples. Secondly, compared to rice, there is no expansion, the numbers are similar - 82 and 84. Normally “expansion” is claimed when an ancestral species has lower number of genes in a family. In that case, the increased number of genes in the family might be indicative of an increased importance of this family for evolution of the new species. No hypothesis was suggested in the manuscript.

Lane 27 “FAR1 genes” since it is first use of the abbreviation it should be spelled out, far-red–impaired response 1.

Line 57-58 Contradictive statements “56% of the estimated genome size is anchored at the chromosome level [2,5]. The MD2 v1 genome that represents the most commonly grown pineapple variety is not anchored at the chromosome level.

Line 81 Please, specify the origin of the MD2 plants. Where from the plants come to the greenhouse in the North Carolina Research Campus. Directly from Hawaii Research Station or from somewhere else.

Line 537 Quite weak justification for characterizing a family.

Line 554 Seven groups. Is this a new grouping or groups were identified previously for FAR proteins from other species? Since no reference was provided, I assume it is a new classification. Then groups must be characterized more detailed. Like a diagram representing domains should be provided with discussion on possible functional significance, since domains are different.

Line 558 Orthologous gene is a gene in different species that evolved from a common ancestor. Very often it is impossible to establish orthology of specific genes in the family, therefore the genes are combined into orthogroups. But orthogroups only make sense when they are established to have related genes in different species. How can an orthogroup be in a single species (sorghum)? To claim orthology you need at least two species. Frankly, having 80 genes in an orthogroup also does not add much to the characterization. Orthology of the genes is important when it provides us with a hypothesis on a function of the new genes. For instance, if a gene has an established role in Arabidopsis and we identified its ortholog in pineapple, we hypothesize that this gene might have similar role in pineapples. If we have 80 orthologs what could we possibly predict about their roles? 

Author Response

Thank you, we appreciate the reviewer comments and edits. It helped us to improve the quality of the manuscript. We addressed all the comments and edits. Please see below a point by point response.

Reviewer 2 Comments
The manuscript describes more complete and better annotated genome of MD2 pineapples, an agriculturally important cultivar. It also made an attempt to characterize one of gene families, which was found to have about 80 members. This was an interesting observation, however, not unique. Authors mentioned that rice also had similar number of genes, while other species varied in the number. It is not clear from the manuscript why authors decided to characterize this particular family. Looks like the only reason was the large number of genes in it. While the first part, the full genome sequencing and better annotation compared to the previous versions of pineapple genome, has clear scientific value and is informative, the second part, characterization of the FAR gene family, is unnecessary, incomplete and somewhat controversial. I would recommend a major revision.

Points to address:
Comment 1. Title “FAR1 gene family” should be “FHY3/FAR1 gene family” or better possibly “FRS/FRF family” as more general and newer name. On the other hand, the use of the term “expansion” for this family claimed in the title is a bit vague. Since it is not clear from the manuscript why the expansion is even claimed. Authors stated that pineapple and rice have similar numbers of FHY3/FAR1 genes (around 80), while other tested species had less (6 in banana and 49 in sorghum).
Answer: The label FAR1 family was changed to FRS/FRF gene family throughout the manuscript (main text, figure and tables). Relative to the expansion we understand the point made by the reviewer, indeed the number of genes is not expanded in pineapple, it is expanded in pineapple and rice. We have changed the FRS/FRF gene family sections to reflect that the increase in gene number occurred in both pineapple and rice and re-modulated the rational to evaluate this family (line 540-559). Also the word “expanded” was changed in the title to reflect more closely the outcome of this analysis.

Comment 2. Lane 26 “Compared with other species genomes, MD2 has an expansion” is incorrectly worded comparison. It reads that MD2 is a species, while MD2 is a cultivar of pineapples. Secondly, compared to rice, there is no expansion, the numbers are similar - 82 and 84. Normally “expansion” is claimed when an ancestral species has lower number of genes in a family. In that case, the increased number of genes in the family might be indicative of an increased importance of this family for evolution of the new species. No hypothesis was suggested in the manuscript.
Answer: The reviewer is correct about MD2 not being its own species. We have changed MD2 to pineapple (see line 27). Compared to dicots and other monocots, there was a larger number of FRS/FRF family genes predicted in both pineapple and rice (Table S11). We made major edit to this section. In the revised section we specified “Due to the large number of FRS/FRF genes in pineapple and rice, we wanted to investigate to what degree the expansion of this gene family was the result of shared duplications vs lineage-specific duplications. Also we wanted to investigate which specific sub-groups expanded, their potential function by orthology, and their possible functional specializa-tion. In the model species Arabidopsis, the FRS/FRF family has been implicated in multi-ple developmental functions including circadian clock regulation and flowering time regulation [10–13], that are traits of particular importance for pineapple cultivar develop-ment. Recent work in pineapple identified some FAR1 genes as been implicated in flower ovule developemnt” (see line 549-559). Based on comparative analysis and mode of duplications we identified the orthogroups (corrected analysis see answer to comment 8) that were expanded in pineapple and based on shared ancestry with functional characterized genes we speculated that these genes may have acquired new specialized important functions in pineapple (line 595-613). Also the impact of the duplications was also highlithed in the last paragraph of section 3.6 (see line 668-676)

Comment 3. Lane 27 “FAR1 genes” since it is first use of the abbreviation it should be spelled out, farred–impaired response 1.
Answer: This issue has been fixed, FAR1 was replaced with FRS and abbreviation added (see line 27).

Comment 4. Line 57-58 Contradictive statements “56% of the estimated genome size is anchored at the chromosome level [2,5]. The MD2 v1 genome that represents the most commonly grown pineapple variety is not anchored at the chromosome level.
Answer: This issue was fixed. The sentences was reworded (see line 56-59).

Comment 5. Line 81 Please, specify the origin of the MD2 plants. Where from the plants come to the greenhouse in the North Carolina Research Campus. Directly from Hawaii Research Station or from somewhere else.
Answer: Added a part that says these plants were shipped from Dole in La Ceiba, Honduras (see line 85-86).

Comment 6. Line 537 Quite weak justification for characterizing a family.
Answer: Taking the reviewer constructive comments into consideration, we made major modifications to the section 3.6 reinforcing the justification for characterizing this gene family (see line 540-559).

Comment 7. Line 554 Seven groups. Is this a new grouping or groups were identified previously for FAR proteins from other species? Since no reference was provided, I assume it is a new classification. Then groups must be characterized more detailed. Like a diagram representing domains should be provided with discussion on possible functional significance, since domains are different.
Answer: These are new groups based on domain content identified AcFARs. We provided further information on the domain content of the groups and the implications of different domain content (see line 571-593, Table S15 column C). We have also added a new supplementary figure with a diagram illustrating the different domains (Figure S14).

Comment 8. Line 558 Orthologous gene is a gene in different species that evolved from a common ancestor. Very often it is impossible to establish orthology of specific genes in the family, therefore the genes are combined into orthogroups. But orthogroups only make sense when they are established to have related genes in different species. How can an orthogroup be in a single species (sorghum)? To claim orthology you need at least two species. Frankly, having 80 genes in an orthogroup also does not add much to the characterization. Orthology of the genes is important when it provides us with a hypothesis on a function of the new genes. For instance, if a gene has an established role in Arabidopsis and we identified its ortholog in pineapple, we hypothesize that this gene might have similar role in pineapples. If we have 80 orthologs what could we possibly predict about their roles?
Answer: Thank you for noting this issue. We realized that there was an issue with the file when parsing the results. All orthologous analysis results have now been corrected. Upon fixing the issue, the results make more sense and are more in line with the results of the phylogenetic analysis. Substantial edits were made in this section (see line 595-613). Multiple orthogroups including functionally characterized genes were identified, and we highlighted their potential similar role in pineapple (see line 605-611).

Round 2

Reviewer 2 Report

The manuscript was greatly improved.